# Deciphering the Importance of Glycosphingolipids on Cellular and Molecular Mechanisms Associated with Epithelial-to-Mesenchymal Transition in Cancer

**DOI:** 10.3390/biom11010062

**Published:** 2021-01-06

**Authors:** Cécile Cumin, Yen-Lin Huang, Arun Everest-Dass, Francis Jacob

**Affiliations:** 1Ovarian Cancer Research, Department of Biomedicine, University Hospital Basel, University of Basel, 4031 Basel, Switzerland; cecile.cumin@unibas.ch (C.C.); y.huang@unibas.ch (Y.-L.H.); 2Institute for Glycomics, Griffith University, Gold-Coast, QLD 4215, Australia; a.everest-dass@griffith.edu.au

**Keywords:** epithelial-to-mesenchymal transition, (EMT/MET), glycosphingolipids, gangliosides, globosides, cell migration/invasion, tumor growth, signal transduction, metastasis

## Abstract

Every living cell is covered with a dense and complex layer of glycans on the cell surface, which have important functions in the interaction between cells and their environment. Glycosphingolipids (GSLs) are glycans linked to lipid molecules that together with sphingolipids, sterols, and proteins form plasma membrane lipid rafts that contribute to membrane integrity and provide specific recognition sites. GSLs are subdivided into three major series (globo-, ganglio-, and neolacto-series) and are synthesized in a non-template driven process by enzymes localized in the ER and Golgi apparatus. Altered glycosylation of lipids are known to be involved in tumor development and metastasis. Metastasis is frequently linked with reversible epithelial-to-mesenchymal transition (EMT), a process involved in tumor progression, and the formation of new distant metastatic sites (mesenchymal-to-epithelial transition or MET). On a single cell basis, cancer cells lose their epithelial features to gain mesenchymal characteristics via mechanisms influenced by the composition of the GSLs on the cell surface. Here, we summarize the literature on GSLs in the context of reversible and cancer-associated EMT and discuss how the modification of GSLs at the cell surface may promote this process.

## 1. Introduction

Epithelial-to-mesenchymal transition (EMT) is a highly dynamic and reversible process where epithelial cells gradually convert into mesenchymal cells. This transition was described for the first time by E.D Hay in 1967 as an important step during gastrulation [1]. In the course of embryonic development, EMT is considered as a critical process for the generation of the mesoderm and endoderm from the ectoderm [2,3]. In adults, similar processes are observed as a physiological response to heal epithelial tissues [4]. In 1982, Greenburg and Hay demonstrated that epithelial cells suspended within a gelling solution composed of collagen I lose their apical–basal polarity, become elongated, and detach from the collagen to migrate as individual cells [5]. Although a well-known mechanism during development, EMT is also required during the invasion–metastasis cascade where cancer cells of the primary tumor gain mesenchymal traits in order to spread and form new and distant colonies.

EMT is not a dual process, but a network of multiple states [6] where tumor cells alter their epithelial and mesenchymal markers, allowing them to display an array of migratory behaviors [7]. This cellular plasticity between both phenotypes is critical for cancer spread, as schematically represented in Figure 1. The metastatic process requires the loss of cell–cell adhesion in epithelial cancer cells through the disruption of the integral membrane proteins responsible for establishing and maintaining epithelial cell polarity [8,9], such as occludin (*OCLN*) [10], claudin 1 (*CLDN1*) [11], and E-cadherin (*CDH1*) [12]. In parallel, the degradation of the basement membrane and extracellular matrix must occur to allow the cells to migrate and invade surrounding tissues. The downregulation of membrane proteins leading to EMT is accompanied by the activation of EMT-inducing transcription factors (EMT-TFs), such as Snail [13], Twist [14], Slug, and ZEB1-2 [8,15]. Here, Snail recruits specific chromatin-modifying enzymes through the DNA-binding motif SNAG to exert transcriptional repression of E-cadherin expression [16]. Similarly, Twist silences E-cadherin expression through binding to the E-box motif at the promoter upstream of *CDH1* [17]. Finally, Slug promotes EMT through the formation of β-catenin complexes, inducing the activation of TGF-β3 [18]. Together with the repression of epithelial markers, EMT also requires activation of mesenchymal markers such as N-cadherin (*CDH2*) [12], vimentin (*VIM*) [19,20] or desmin (*DES*) [21] for the cells to gain motility [22,23].

Following the dissociation of the primary tumor, cancer cells invade the endothelial lamina for intravasation into the vascular system to extravasate and colonize a new metastatic site. In the new environment, single tumor cells usually undergo mesenchymal-to-epithelial transition (MET). MET is suggested to be the reverse process of EMT where tumor cells gain back their epithelial features and lose their mesenchymal phenotype. The induction of a mesenchymal tumor into an epithelial non-stem-like state has been suggested to occur during carcinoma progression and tumor outgrowth at distant sites [24,25]. Interestingly, the primary tumor and metastatic sites present molecular and morphological similarities at the histopathological level but the associated pathways between EMT and MET are not identical [26]. Thus, various studies investigated the cell modifications that could induce EMT.

Pattabiraman and co-authors treated human mammary epithelial cells using cholera toxin, a multimeric protein complex from *Vibrio cholerae* known to bind a major ganglioside-type glycolipid called GM1 [15]. The cholera toxin treatment induced the activation of the protein kinase A signaling pathway, triggering mesenchymal cancer stem cells to gain epithelial stem-like properties. Furthermore, it is known that alteration of glycosylation on lipids at the cell surface can affect cell adhesion, recognition, and signal transduction [27,28] through the regulation of different signaling pathways involved during EMT. The EMT-associated alteration of glycosphingolipids (GSLs) has been identified in several cancer types; e.g., reduced globosides and enhanced gangliosides in mesenchymal-like PaTu-T cells compared to epithelial-like PaTu-S cells have been detected using mass spectrometry-based glycomics and flow cytometry [29], the functional loss of globosides induces EMT in ovarian cancer using the CRISPR-*Cas9* approach [30], or during transition of breast cancer stem cells [31]. In addition, the emerging role of GSLs in epithelial-to-mesenchymal transition (EMT) was previously mentioned as a process that enables metastatic cellular invasion in the context of cancer progression, highlighting the spontaneous and transforming growth factor β (TGFβ)-induced alterations of GSLs [32]. Thus, we have considered the literature reporting on GSLs and reversible EMT-associated cancer cell mechanisms, such as cell migration and invasion, drug resistance, or proliferation, to be eligible for this review (highlighted in Figure 1) [33]. The review summarizes the literature on GSLs in the context of EMT/MET with a specific focus on cancer and discusses how the modification of GSLs may trigger this transition.

## 2. Human Glycosphingolipids Biosynthesis

Glycosphingolipids (GSLs) are complex glycan structures linked to a ceramide backbone by a β-glycosidic linkage [34] and are the most represented class of glycolipids in vertebrates [35]. The number of glycans (comprised of ~20 monosaccharide residues) attached to the ceramide-forming GSLs is predicted to be more than 400 different unique structures [34]. Their synthesis is initiated at the cytosolic membrane of the endoplasmic reticulum (ER) where the ceramide is produced, and then is converted into galactosylceramide or glucosylceramide (GlcCer) in the ER or *cis*-Golgi, respectively [32,36]. GlcCer GSLs are the major class of GSLs described in the literature in the context of cancer, and thus the main class highlighted in this review. GlcCer is further galactosylated to form lactosylceramide (LacCer), the major glycan substrate for branching into other GSL series [37,38]. The addition of specific monosaccharides to the precursor LacCer further categorizes GlcCer-GSLs into seven series, namely, the ganglio-, lacto-, neolacto-, globo-, isoglobo-, mollu-, and arthro-series. Of these, the major series described in the literature in vertebrates are the ganglio-, globo-, and neolacto-series (Figure 2), each of which follow a specific biosynthetic pathway described below.

The synthesis of globosides starts with the transfer of galactose from UDP-Gal to LacCer by the Gb3 synthase (Gb3S encoded by *A4GALT*) to form globotriaosylceramide (Gb3), also referred to as CD77 or Pk. The synthesis is followed by the addition of *N*-acetylgalactosamine through the enzyme β1,3-*N*-acetylgalactosamintransferase 1 (*B3GALNT1*), resulting in Gb4, the precursor for SSEA3 (Gb5), which is synthesized by the enzyme β1,3-galactosyltransferase 5 (*B3GALT5*). Finally, SSEA3 is converted to either SSEA4 by β-galactoside α2,3-sialyltransferase (*ST3GAL2*) or to GloboH by α1,2-fucosyltransferase (encoded either by *FUT1* or *FUT2*). The Gb3 and Gb4 GSLs constitute the basis of the P-blood group system while Gb5 (SSEA3) and SSEA4 are cell-surface markers used to define human embryonic stem cells [39] and have been associated with several malignant diseases [40,41].

For progression of LacCer into the ganglioside-series pathway, the first ganglioside, GM3, is synthesized by the addition of a sialic acid to LacCer from a CMP-sialic acid donor. The initial transfer of this first sialic acid is synthesized by LacCer α2-3 sialyltransferase (GM3 synthase or “GM3S”, encoded by *ST3GAL5*), with further extension by different α2-8 sialyltransferases (GD3 synthase or ‘GD3S’ encoded by *ST8SIA1*) and GT3 synthase (GT3S encoded by *ST8SIA5*). Elaboration of gangliosides with *N*-acetylgalactosamine to form GM2, GD2, and GT2 is catalyzed by B4GALNT1, while the subsequent synthesis of GM1, GD1b, and GT1c are is catalyzed by B3GALT4 through the addition of galactose. Finally, the synthesis of GD1a and GT1b, via the addition of a sialic acid residue to the terminal galactose, is catalyzed by ST3GAL1 [42]. The third vertebrate series comprises (neo-) lactosides, starting with Lc3 as the major precursor that is synthesized by the addition of a GlcNAc residue to LacCer by the enzyme B3GNT5. The formation of two branches for the synthesis of either Lc4 or nLc4 is catalyzed by the enzymes B3GALT5 and B3GALT1, respectively. Moreover, both Lc4 and nLc4 can be further elongated through the addition of fucose to form Lewis structures called the Lewis blood group antigens.

## 3. Globo-Series GSLs

During EMT, modification of GSL expression at the cell surface can alter the receptor tyrosine kinase (RTKs) downstream signaling pathways. Several studies have reported on GSLs functional roles by analyzing the different EMT hallmarks, such as proliferation, migration/invasion, and their tumorigenesis, but also how the cellular signaling pathways convert immotile and polarized epithelial cells into migratory and invasive cells [43].

### 3.1. Gb3 and Gb4

A comparison of colorectal cancer samples and their corresponding controls using MALDI-TOF-MS and immunofluorescence microscopy identified the shortest globoside, Gb3, as a marker in the tumor tissue of colorectal cancer and their metastases [44,45,46]. Interestingly, an EMT scoring established by Tan et al. associated colorectal cancer as predominantly an epithelial cancer type [47], which is in line with other studies [46,48]. Patient-derived human breast cancer xenografts and human breast tumor with high Gb3 expression were treated using Shiga toxin, an oligomeric complex that binds specifically to Gb3 at the cell surface, showing that Gb3 was preferentially accumulated in epithelial and endothelial cells of the tumor [49,50,51], coinciding with the expression of epithelial markers.

The prominent expression of Gb3 in epithelial cancers has led to various Gb3-targeting treatment attempts. For instance, knockdown of Gb3S (encoded by *A4GALT*), using siRNA in colon cancer epithelial cells [46] or Shiga toxin in patient-derived gastric adenocarcinomas [52], decreased the cells’ capacity to migrate and proliferate in vitro and in vivo compared to cells with impaired Gb3 expression [52]. Gb3S is required for synthesis of all globosides; Desselle et al. generated a mouse IgM mAb specific for Gb3 to treat endothelial cells (HMEC-1). By targeting Gb3, a globoside that is associated with pro-angiogenic features, the authors inhibited angiogenesis in vitro and ex vivo in HMEC-1, as well as cell proliferation to block metastatic spreading [53]. Ovarian carcinoma cell lines also express Gb3 [54,55,56]; we have previously demonstrated that the depletion of globosides in ovarian cancer cells using CRISPR-*Cas9* resulted in a loss of epithelial markers and induced an EMT accompanied with a reduction in cell proliferation and enhanced cell motility both in vitro and in vivo [30]. Interestingly, loss of globosides was accompanied with an increase of ganglioside GM1 shown by flow cytometry and mass spectrometry. The switching of the core structures of the GSLs has also been described as a consequence of embryonic stem cell differentiation [57,58], suggesting that specific GSL structures are markers of differentiation, including cancer-associated EMT/MET (summarized in Table 1).

Induction of EMT through the modification of globosides at the cell surface requires the modulation of internal cell signaling. Globosides alone may induce the activation of ERK signaling and p38 MAPK without any additional growth factors. Enzymatic inactivation of GlcCer GSL synthesis using the GlcCer synthase inhibitor EtDO-P4 in colon (HCT116) and breast (MCF7) cancer cell lines suppressed activation of the epidermal growth factor (EGFR)-inducing ERK pathway and various RTKs. Only exogenous addition of Gb4 could restore the activation of ERK, showing direct interaction with EGFR [59], as represented in Figure 3. Of note, GlcCer synthase inhibitors are supposed to inhibit all GlcCer and our work in various cancer cell lines have indicated that the decrease in the expression of one or more particular GSLs is dependent on the cell line under investigation, the type of inhibitor applied, and the exposure conditions [56]. Another study in human ovarian carcinoma demonstrated that overexpression of Gb3 increases doxorubicin drug resistance in human ovarian carcinoma through the activation of cSrc kinase, decreased β-catenin phosphorylation, and increased nuclear β-catenin [60] (Figure 3). The anthracycline antibiotic substance doxorubicin elicits cytotoxic effects by intercalating into DNA and has been described in the context of acquired drug resistance as part of EMT [30,61]. These studies demonstrated that the globoside Gb3 on the cell surface of primarily epithelial cancer cells are involved in EMT and that either enzymatic inhibition or gene deletion of A4GALT induces a loss of epithelial features in various cancer types.

### 3.2. SSEA3, SSEA4, and GloboH

Apart from Gb3 and Gb4, expression of longer globosides (stage-specific embryonic antigens SSEA3, SSEA4, and Globo H) has been described in various cancers and EMT. Several studies independent of the cancer type demonstrated that globosides seem to be linked to an enhanced cell capacity to proliferate in vitro and in vivo, a key step in the formation of primary tumors as well as metastatic outgrowth [63]. Cheung et al. demonstrated that elevated SSEA3 in CD44^+^/CD24^−^ breast cancer cells leads to enhanced tumor growth in vivo and therefore suggested SSEA3 as a breast cancer stem cell marker [40]. After deriving fallopian tube epithelial cells from the epithelial monolayer of the human fallopian tube epithelium, Chang et al. observed a formation of spheroids in gelatin-coated cultures. Downstream analysis of these spheroids revealed expression of SSEA3 and SSEA4 [64]. Abundant SSEA4 expression was also noted in the epithelium of benign serous cystadenoma with a gradual decrease towards advanced tumor stage [65].

Targeting SSEA3 and SSEA4 using antibodies or ET-18-Me in mammary carcinoma cells altered the globoside expression and increased cell motility and invasiveness (MCF-7/AZ) [66,67]. *B3GALT5* knockdown also leads to apoptosis in cancer cells with marginal effect in normal cells [40]. It was also shown in a study that *B3GALT5* silencing mediates apoptosis through the dissociation of the RIP from FAK complex, thereby further being associated with the Fas-associated protein with death domain (FADD), and then facilitating the activation of caspase 8 and 3 and FAK degradation [68]. Interestingly, other studies have also associated globosides as a marker for tumorigenesis. Kuo et al. observed SSEA4, SSEA3, and GloboH expression only in hepatocellular carcinoma (HCC) tumor tissues but not in normal tissues, using immunohistochemistry and mass-spectrometry. Following an examination of the gene expression of their biosynthetic enzymes, *FUT1*, *FUT2*, *B3GALT5*, and *ST3GAL2* in hepatocellular carcinoma tissue samples, using RT-qPCR, revealed an association of *FUT1* or *B3GALT5* expression with advanced stages [69]. In line, *FUT1* and *B3GALT5* appeared as promising genes associated with the GSLs investigated in this hepatocellular carcinoma study and were further suggested as specific targets for therapeutic intervention. The distinct expression of SSEA3 and SSEA4 was also identified in colorectal cancer cell lines (Caco-2, DLD-1, HT-29, SW480, and HCT116) known to express various epithelial markers. Using fluorescence-activated cell sorting, Sivasubramaniyan et al. sorted prostate cancer cells with high expression for SSEA4 and observed a reduction in the epithelial markers Claudin-7 and E-cadherin, accompanied by an increase in tumorigenicity in mice (in vivo). In parallel, they observed that the same cell line with a lower expression for SSEA4 formed cobblestone-like epithelial colonies [62]. A similar conclusion has been reported by Suzuki et al. in colorectal cancer where SSEA3 was used as the mesenchymal stem cell marker [70]. Van Slambrouck et al. suggested the signaling pathway behind SSEA4 and the cell’s capacity to invade. Immunoprecipitation of SSEA4 revealed an association with cSrc (phosphor-Tyr^416^) and FAK (phosphor-Tyr^397^) in breast cancer cells. Phosphorylation of amino acid residues T308 and S473 by PDK1 and mTORC2, respectively, is essential for full AKT activation and promotes the inactivation of Rheb, a RAS family protein that activates mTORC1. These processes induce the phosphorylation of ribosomal protein S6 kinase (S6K) and the eukaryotic translation initiation factor 4E-binding protein 1 (4E-BP1), releasing downstream the eukaryotic translation initiation factor 4E (eIF4E). Both, S6K and eIF4E affect protein translation and cell proliferation through the transduction of mitogen and nutriment signals [71]. Other studies also demonstrated that Akt phosphorylation leads to activation of most notably glycogen synthase kinase 3 (GSK3), tuberous sclerosis 2 (TSC2), caspase 9, and PRAS40 (AKT1S1), which might explain its relatively wide spectrum of downstream effects in promoting cell proliferation, differentiation, apoptosis, angiogenesis, and metabolism [72]. Interestingly, neither the globosides Gb3 and SSEA3 nor the ganglioside GM2 were capable of triggering this signaling pathway in the same model [67]. In SSEA4-enriched prostate cancer cells, the authors reported on an association between the signaling nodes pSrc, pAKT, and pPI3K [62].

## 4. Ganglio-Series GSLs

Gangliosides can induce the activation of different EMT hallmarks through the regulation of signaling pathways. As with globosides, gangliosides have also been reported to have a positive or negative retro-control of the different signaling pathways. There seems to be a correlation with the number of sialic acids (mono- or di-sialylated) and the role of gangliosides during EMT, with some conflicting uncertainties [73,74]. Different studies will be reviewed in this chapter, highlighting the multifaceted roles of gangliosides for the maintenance of cell homeostasis and for the positive or negative regulation of the malignant properties of cancer cells.

### 4.1. Monosialylated Gangliosides

TGF-β signaling has been shown to play an important role in tumor development and is commonly used to induce EMT in various epithelial cells [75]. Initiation of EMT through TGF-β1 in human lens epithelial cells HLE B-3 increases the expression of the ganglioside GM3, correlating with *ST3GAL5* expression [76], the key enzyme for the synthesis of most complex gangliosides. In order to elucidate the action of this ganglioside during EMT, different studies have silenced *ST3GAL5* using siRNA. In murine breast cancer cells (4T1), Gu et al. observed a reduction in cell migration, invasion, and anchorage-independent growth in vitro as well as lung metastasis in vivo [77]. It was demonstrated that gene silencing of ST3GAL5 enhances phosphorylation of FGFR, which activates the PI3K/AKT pathway and resulted in increased Akt phosphorylation at Thr308 and Ser473 in addition to the inhibition of PTEN phosphorylation [77] (Figure 4). The PI3K–Akt pathways are important in regulating cell proliferation and maintaining the cellular characteristics of malignant cells [78,79]. These data suggest that ST3GAL5 may trigger the transition of epithelial and mesenchymal cells in various cancer types. However, ST3GAL5 synthesizes various ganglioside GSLs, usually characterized by the number of sialic acids attached to it. For this reason, complementary studies investigated the impact of GM3 and GM1 alone in cancer development.

Constitutive GM3 knockdown in human lens epithelial cells also suppresses cell migration through the interaction of GM3 with the TGF-β receptor [76]. Exogenous addition of GM3 to HCT116 colon cancer cells have an anti-proliferative effect by stimulating cyclin-dependent kinase inhibitor (CDKI) p21 (WAF1) through the inhibition of the PI3K/AKT/MDM2 survival signaling pathway [80] (Figure 4). It is also known that exogenous addition of GM3 and GM1 to human epidermoid carcinoma (KB and A481) cell lines inhibited cell growth through the downregulation of EGF-stimulated tyrosine phosphorylation [84]. A similar mechanism of action was observed after treatment with GM1 and GM3 in the human neuroblastoma cell line NBL-W, inhibiting EGFR phosphorylation [85]. Another study showed that the interaction of GM1 with platelet-derived growth factors (PDGF) reduces PDGF receptor phosphorylation and reduced the activation of MAPK, which resulted in reduced cell proliferation [86]. Together, these publications provide evidence that the monosialylated ganglioside GM3 and GM1 are involved in metastatic processes and seem to have a pro-tumoral action. Fujimoto et al. further confirmed this by exogenously adding GM3 and GM1 in glioma and inducing inhibition of cell proliferation [85]. However, some studies have also shown mono-sialylated gangliosides to take a protective role in the context of cancer cells, as described below.

TGF-β treatment of human epithelial cells HCV29 and NMuMG induced EMT, leading to a reduction in GM2, but not in GM1 or GD1a, on the cell surface, using HPTLC [87,88]. Here, the addition of exogenous GM2 abrogated the cell motility, which was not the case for GM1 or GM3 [88]. Previous publications demonstrated that GM3 and GM1 overexpression were in favor of tumorigenicity. However, in rat PC12 cells, GM1, which is directly associated with Trk, the high-affinity tyrosine kinase-type receptor for nerve growth factor (NGF), strongly enhances neurite outgrowth [89]. Moreover, treatment against GM3 on the human neuroblastoma cell line (NBL-W) inhibited cell proliferation [90]. In the same way, a GM3 knockout mouse embryonic fibroblast promoted cell migration through the activation of Ras/Raf/MEK/ERK signaling [91]. Finally, other evidence demonstrated that addition of exogenous GM3 suppressed the VEGF-induced migration of human umbilical vein endothelial cells in vitro and in vivo [92].

Similarly, genetic repression of the transcription factor ZEB1 is usually upregulated during EMT and increases ganglioside expression through ZEB1-mediated ST3GAL5 activation in mammary epithelial cells (NM18) and is accompanied with an increase in intercellular adhesion [93,94]. These observations were confirmed in human normal bladder HCV29 and human mammary carcinoma MCF7 cells treated with EtDO-P4, enhancing cell motility [88]. However, researchers have to be cautious in using these GlcCer inhibitors, as they have been reported to be selectively acting on GSLs and in a cell line-dependent manner [56].

These data together suggest that GM3 has a prominent role during EMT. However, there is also evidence that the metastatic capacity can be due to the expression of more complex gangliosides as a consequence of the dynamic alterations of GSLs upon experimental alteration. For these reasons, specific aspects of these two gangliosides need to be characterized in more detail and should be considered in a broader context with GSLs of different series.

### 4.2. Disialylated Gangliosides

The association of di-sialylated gangliosides with tumorigenicity and during EMT has been described primarily for GD3 and GD2. Both GSLs were suggested as potential breast cancer stem cell markers mainly involved in pro-tumoral features in cancers through the regulation of various signaling pathways.

Studies have also associated GD3 synthase (ST8SIA1), which is the key enzyme for GD3, GD2, and GD1 synthesis, with EMT. Sarkar et al. demonstrated that ST8SIA1 is important for the initiation and the maintenance of EMT. Knockdown of *ST8SIA1* with RNA interference in MDA-MB-231 reduced the cell’s capacity to invade and form metastasis in vivo [81]. These data were further supported by Battula et al., demonstrating that silencing of *ST8SIA1* reduced the cell proliferation and decreased mammosphere formation in vivo [95]. The described mechanistic consequence upon *ST8SIA1* silencing was linked to reduced phosphorylation of c-Met Tyr1234/1235 as well as its downstream signal mediator phosphor-Akt [81]. Finally, introduction of ST8SIA1 in the osteosarcomas cell lines increased GD3 and GD2 expression and improved the cell’s capacity to migrate and invade [96]. An increase in ST8SIA1 in rat pheochromocytoma cell line PC12 induced phosphorylation of Trka and the activation of ERK1/2 through the Ras/MEK/ERK pathways, inducing the enhancement of the proliferation capacity of these cell lines [82].

GD2 was described to be elevated in triple-negative breast cancer stem-like cells. CRISPR-knockout of *ST8SIA1*, the key enzyme of GD3, GD2, and GD1 expression in these breast cancer stem-like cells, inhibited metastasis and mammosphere formation in vivo [83]. Nguyen et al. demonstrated that GD2+ breast cancer stem cells activate FAK–AKT–ERK–mTOR signaling, with an upregulation of FAK and the downstream proteins Csk, 4E-BP1, and STAT3. In the same GD2+ breast cancer cells, FAK and 4E-BP1 were more phosphorylated compared to the GD2 cells, suggesting that FAK–AKT–ERK–mTOR signaling is regulated by *ST8SIA1* [83,96]. An association between elevated GD2 levels and an increasing histopathological grade in bladder cancer was also reported by Vantaku et al., using different flow-cytometry analyses; they demonstrated a muscle invasive cell line UMUC3 with elevated GD2 expression, showing a cancer stem cell property (CD44^+^/CD24^−^) as well as a mesenchymal phenotype. Moreover, the same GD2+ cells displayed higher expression for vimentin and lower E-cadherin expression than the GD2^−^ cells [97]. Battula et al. observed a similar pattern in GD2-positive breast cancer stem cells, with an association between the CD44^+^ and CD24^−^ stem cell markers and generation of spheroids in vitro [95]. Elevated GD3 and GD2 were also described in small-cell lung cancer, promoting enhanced proliferation as well as invasion activities [98]. Proliferation in MDA-MB-231 cells requires GD2 expression via the constitutive activation of cMet. The accumulation of GD2 and cMet could reinforce the tumorigenicity and aggressiveness of breast cancer [99]. PI3K/AKT and c-Met receptors were shown to be regulated by GD3 in breast cancer [100].

## 5. An Overview of Current Methodologies to Characterize GSLs

Since the advances in purification and analytical methods, hundreds of different GSL glycan species have been annotated in humans, and their expression is found to be highly dynamic due to physiological and environmental conditions [119]. Owing to the amphipathic nature of GSLs, consisting of a hydrophobic lipid and hydrophilic glycan moiety, early approaches such as thin-layer chromatography (TLC) have been usefully applied to monitor the purity and quantity of extracted GSLs. Several methods have been developed to visualize the GSLs after TLC separation, including staining with orcinol or diphenylamine and *p*-anisaldehyde, ninhydrin, and thymol-sulfuric solution for the glycolipids and glycans, respectively [120]. Furthermore, TLC-resolved GSLs were also combined with immunostaining procedures using GSL-recognizing antibodies, lectins, or bacterial toxins to determine the GSL interacting receptors, or to define antibody specificity [121]. More frequently, many studies have applied the antibody-based immunohistological staining and flow cytometry analysis [56] of GSL expression both in vitro and in vivo [30]. Using advanced mass spectrometry methods, TLC-resolved GSLs can also be characterized using MALDI mass spectrometry (MS) [122]. MALDI-MS has enabled the sensitive detection of many analytes directly from TLC-plates, even preceding any staining. Since GSLs are glycoconjugates composed of an oligosaccharide linked to a ceramide. The biophysical properties of these ceramides modulating cell membranes vary based on the acyl chain composition, namely, long (C16- and C18-), very long (C24-), and unsaturated (C18:1- and C24:1-), hereby regulating the cancer cell properties [123]. Thus, glycomics combined with untargeted metabolomics and lipidomics using mass-spectrometry may provide further insight into the composition and function of these glycoconjugates [124]. Here, the composition of a fatty acyl and glycan moiety was determined in detail solely by TLC–MALDI-MS, without prior derivatization, enzymatic cleavage, and/or reversed phase separation. Moreover, recent technological advances in analytical platforms have resulted in increased untargeted metabolomic and lipidomic studies, which have revealed alterations in lipid metabolism and immune pathways that directly implicate GSL changes and function [125,126].

Complete structural analysis of GSLs usually requires a combination of techniques to determine the glycan composition, glycosidic linkage, (α or β) anomeric configuration, and the fatty acid chain of the ceramide moiety. Over the past years, nuclear magnetic resonance (NMR) and mass-spectrometry (MS)-based technologies combining liquid chromatography have been useful for structural analysis of GSLs. MS-based technologies, such as electrospray ionization (ESI-MS) [127,128], matrix-assisted laser desorption/ionization (MALDI) [129], as well as ion-mobility MS [130], are frequently utilized in studies investigating intact glycolipids. The structural information obtained by tandem MS is often complemented by NMR or enzymatic digestion to acquire the linkage and anomeric configuration, which are essential to define the full structure. However, the major limitations of these techniques are labor intensive, time-consuming, and cost-ineffective for initial comparison of complex biological samples. Another approach that is commonly used is to enzymatically cleave the glycan headgroups using a broad specificity endoglycoceramidase, followed by purification and detailed characterization of the glycan moiety alone [131,132,133]. A new technical method, mass spectrometry imaging (MSI), enables the molecular imaging of tissue sections using MALDI-MS. The spatial distribution of individual GSLs in tissue sections at a cellular resolution (up to 10 μm resolution) are now possible and are used to stratify cancer biopsies and understand the molecular basis of tumor development and GSL metabolism [134].

A recent development, a lectin microarray, demonstrated a time-saving and low-cost manner to directly detect the glycosylation patterns of GSLs without glycan release from complex biological samples [135]. Moreover, emerging studies, such as multiplexed capillary gel electrophoresis coupled to laser-induced fluorescence (xCGE-LIF) [136] and ultra-performance hydrophilic interaction liquid chromatography with fluorescence detection (UPLC-HILIC-FLD) [132], show the technical potential for automated analysis of GSL-derived glycans. Overall, with the rapid development and evolution of modern technologies during the last years, the comprehensive glycolipidomic profile could provide extensive and valuable insights for future diagnosis and treatment targets for GSL-associated diseases beyond classical single GSL studies. Promising discoveries with regard to GSL biomarker candidates have been made in human liquid biopsies, with recent studies predicting tumor growth and progression. Although most GSL analysis have been carried out from tissues and cells, there have been some interesting studies on the analysis of GSLs from liquid biopsies. Li et al. profiled 49 gangliosides in serum samples from healthy women and patients with either benign breast tumor or cancer, using liquid-chromatography-Fourier transform mass spectrometry (LC-FTMS). Here, the authors reported on a gradual increase in GM1, GM2, GM3, GD1, and GD3 species towards malignant tumors [137]. A similar observation was reported in cholangiocarcinoma, demonstrating that the GM2 concentration was elevated in the serum of patients with cholangiocarcinoma [138]. Elevated gangliosides in cancer patients were also reported in blood samples in 73 children with high-risk neuroblastoma compared with 40 cancer-free children using LC-TMS. In particular, the GD2 median-concentration was 30-fold higher than the median concentration in the control group [128].

We and others have shown [40,55,68,103,131,139] the versatility of using lectin flow cytometry as well as highly sensitive mass spectrometry to characterize and demonstrate altered changes on GSLs. Currently, GSL characterization is mainly performed on bulk analyses of complex tissues or large cell populations, yielding an averaged overview. Cellular heterogeneity is an intrinsic attribute, even for cells originating from the same genome, and have diverse chemical and physiological features due to unique microenvironments and random processes. Astounding improvements in bioanalytical technology will enable the future development of GSL analysis from single cells and demonstrate the vital roles they play in cellular metabolism and processes.

## 6. Conclusions

GSLs are a continuously emerging class of biomolecules with increasing evidence on their importance in patho-physiological processes, including human malignancies. However, despite the large number of GSLs annotated (http://sphingolab.biology.gatech.edu/index.html), only a few GSLs are described in the literature. Our review summarizes the current literature and shows the emerging evidence of GSLs triggering EMT in cancer through multiple alterations in cell signaling pathways. Several studies demonstrated that alteration of GSLs through enzymatic inhibition, gene silencing or editing, and exogenous addition could modulate the cancer signaling pathways, contributing to cell proliferation and motility—classical features of EMT. Interestingly, the expression of globosides are mostly described in cell types of epithelial origin or known to express markers typically associated with epithelial characteristics and tumor formation. Vice versa, gangliosides might be elevated in mesenchymal cell types and associated with elevated cell motility. However, the literature seems to be less consistent for this series. Not to be neglected, the neolacto-series is an important series among all the GSLs and is expected to be involved in EMT, too. It also remains uncertain how these glycans on lipids drive the metastatic dissemination of non-epithelial cancer cells. In this review, we have not addressed the contribution of the ceramide composition, which remains largely unknown considering the lipid and glycan moiety together, but which might also be a direction for future studies. In order to gain a deeper insight into the function of GSLs in cancer and EMT, future studies might need to translate the current findings from indirect measurements using lectins or antibodies and in larger cohorts of human tissue samples, with a special focus on the spatial and cell type composition as well as intra-patient heterogeneity. The development of new methods or adaption of already-established, single-cell approaches, and incorporation of GSL-specific technical pipelines, will become available through the increasing number of and advances in technologies.

## Figures and Tables

**Figure 1 biomolecules-11-00062-f001:**
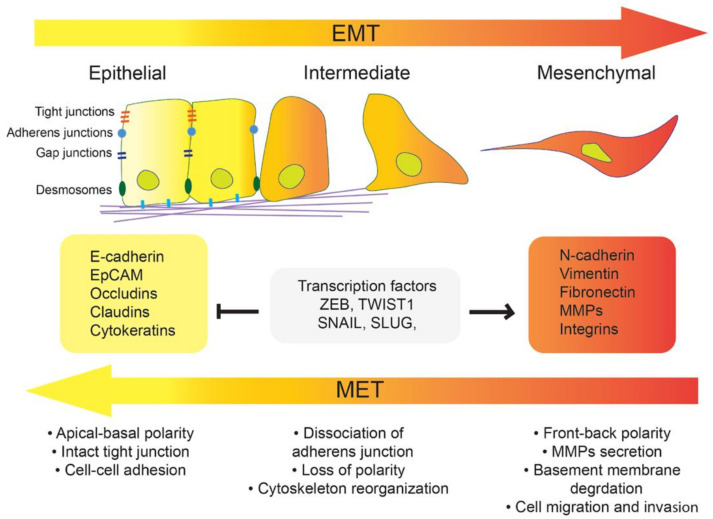
Epithelial-to-mesenchymal transition (EMT) involves the reorganization of the apical–basal polarity, the disruption of tight adherens and gap junctions, as well as the loss of epithelial markers such as E-cadherin, occludin, and claudin. In parallel, cells undergoing EMT detach from each other, secrete matrix metalloproteinases, basement membrane degradation, and acquire migratory and invasive properties. These cells show elevated expression of mesenchymal markers such as N-cadherin, cytoskeleton protein vimentin, and MMPs, which are regulated by various transcription factors (ZEB, TWIST, SNAIL, and SLUG).

**Figure 2 biomolecules-11-00062-f002:**
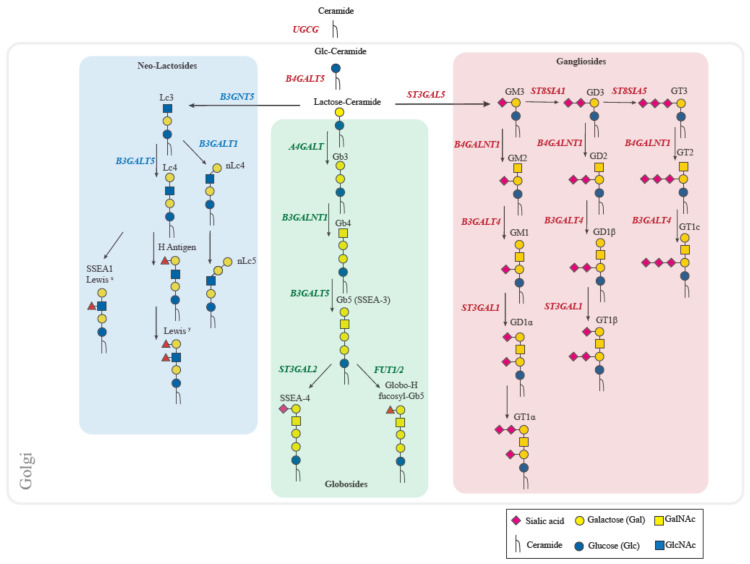
Schematic representations of the GSL synthetic pathways. Ceramide is glucosylated in the *cis*-Golgi and then galactosylated to form lactosylceramide. GSLs are classified into three major series in vertebrates: (neo-) lacto-series (blue), globosides (green), and gangliosides (red). These GSL series are directly synthesized from the lactosylceramide precursor, with distinct glycosyltransferases determining which elongation process is followed: *B3GNT5* for synthesis of the (neo-) lacto series, *A4GALT* for globosides and *ST3GAL5* for gangliosides.

**Figure 3 biomolecules-11-00062-f003:**
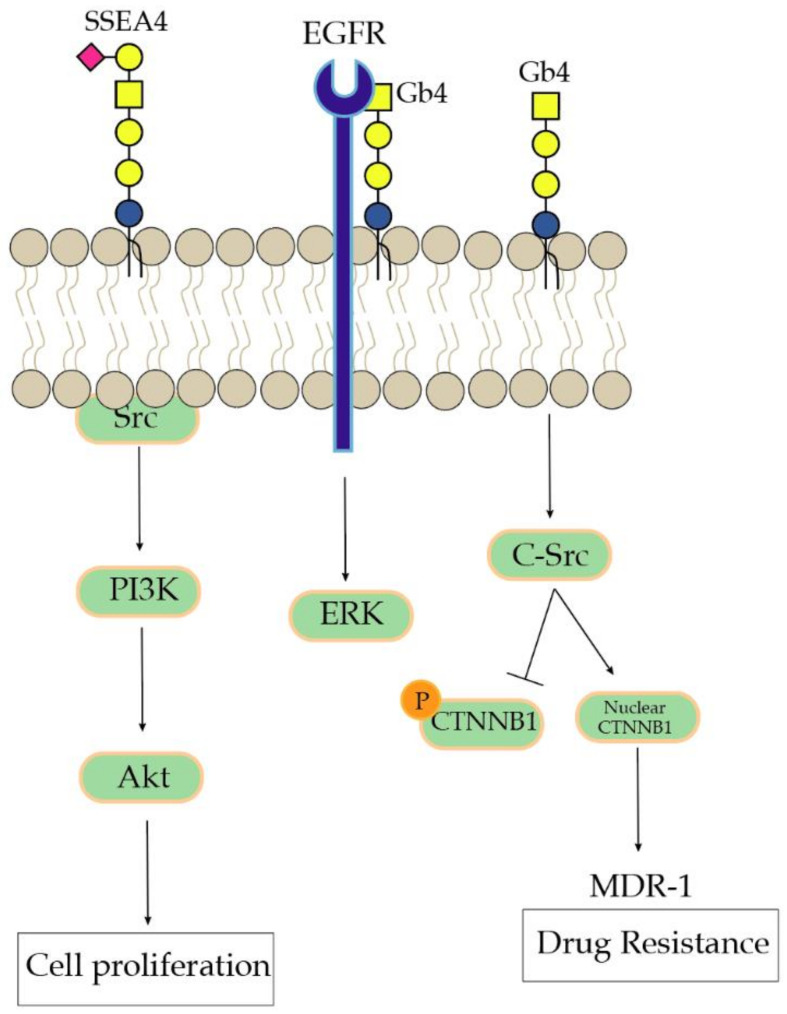
In prostate cancer cells, SSEA4 through the colocalization with activation of pPI3K, pAkt, and pSrc activates cell proliferation [62]. Globoside Gb4 promotes activation of ERK by interaction with the epidermal growth factor receptor in HCT116 and MCF7 [59]. Gb4 on a lipid raft activated cSrc kinase, leading to decreased β-catenin phosphorylation accompanied with increased nuclear β-catenin. Consequently, β-catenin/Tcf4-mediated binding at the MDR1 promoter region activated P-gp expression and induces drug resistance [60].

**Figure 4 biomolecules-11-00062-f004:**
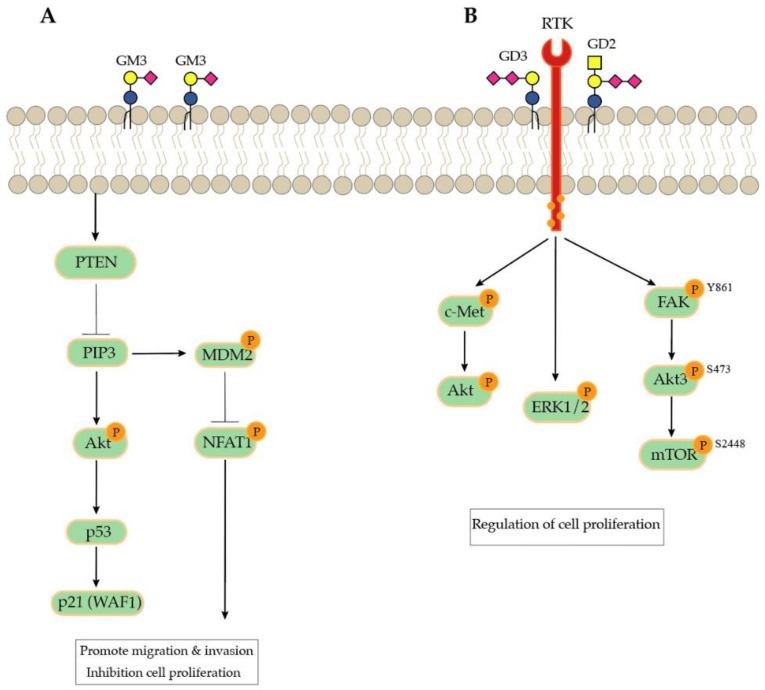
(**A**) PTEN modulates cell survival and proliferation by suppressing PI3K/Akt signaling. *ST3GAL5* silencing reduced ganglioside expression and activated PI3K/Akt signaling inhibits cell migration and invasion [77]. While, the exogenous ganglioside GM3 stabilizes p53, inducing an increase in the p21^WAF1^ protein, involving inhibition of cell proliferation [80]. (**B**) GD3S regulates EMT through the activation of the c-met signaling pathway [81]. In PC12 cells, GD3S continuously activate Trka and ERK1/2 and enhanced proliferation [82]. Finally, in breast cancer stem-like cells, a higher GD2 expression activates the FAK–AKT–mTOR signaling pathway [83].

**Table 1 biomolecules-11-00062-t001:** Summary of the EMT hallmarks associated with GSLs in different cancer types. The table highlights the individual GSLs and their corresponding KEGG annotated genes encoding the glycosyltransferases and synthesizing the specific GSL. The genes/proteins investigated in each study are bold and an asterisk (*) was added when the glycan was not specifically investigated.

	GSL	KEGG	Tumor Type	Associated Cell Characteristics	References
**Globosides**	Gb3	*A4GALT*	Neuroblastoma	Anti-Gb3 antibody inhibits angiogenesis and tumor development	[53]
		***A4GALT***	Ovarian	Promotes EMT, cell migration, chemoresistance and reduces cell proliferation	[30]
		*A4GALT*	Gastric	Gb3 is expressed in gastric adenocarcinoma	[52]
		*A4GALT*	Colon	Upregulated in metastatic colon cancer, promotes cell invasiveness, and tumor growth	[46]
	Gb4	*B3GALNT1*	Colorectal	EtDO-P4 reduces cell proliferation. Exogenous Gb4 activates EGFR and induces the ERK pathway	[59]
	SSEA3 (Gb5)	***B3GALT5***	Breast	Promotes cell proliferation, tumor growth, cancer stemness	[40,65,68]
		*B3GALT5*	Colorectal	High tumorgenicity and promote cell proliferation in vivo	[70]
	GloboH	*FUT1* and *FUT2*	Breast	Promotes cell invasion, reduced apoptosis	[68]
	SSEA4	*ST3GAL1, ST3GAL2*	Glioblastoma	Anti-SSEA4 antibody inhibits tumor growth in vivo	[63]
		*ST3GAL1, ST3GAL2*	Ovarian	Loss of SSEA4 correlated with advanced tumor stage and poor cell differentiation	[65]
		*ST3GAL1, **ST3GAL2***	Breast	SSEA4 promotes chemoresistance, tumorigenicity in vivo and promotes EMT	[101]
		*ST3GAL1, **ST3GAL2***	Prostate	SSEA4-positive cells downregulate epithelial cell-associated markers, promotes EMT and cell–ECM adhesion	[62]
		*ST3GAL1, ST3GAL2*	Oral	Cancer stemness, promote tumorigenicity in vivo	[102]
**Lacto-neolacto**	P1	*A4GALT*	Ovarian	Promotes cell migration	[103]
	Lc3, nLc4	*B3GNT5*	Leukemia	Acute myeloid leukemia initiation and differentiation	[104]
**Gangliosides**	GM3	*ST3GAL5* (*SIAT9*)	Ovarian	Retinoid-resistant ovarian cancer cells have higher level of GM3	[105]
		***ST3GAL5*** ***(SIAT9)***	Breast	*ST3GAL5* silencing inhibits cell migration	[77]
		*ST3GAL5* (*SIAT9*)	Glioblastoma	Exogenous GM3 inhibits proliferation/migration	[90,106]
		***ST3GAL5*** ***(SIAT9)***	Colorectal	Induces cisplatin-induced apoptosis	[107]
		***ST3GAL5*** ***(SIAT9)***	Leukemia	Promotes leukemia cell line differentiation	[108]
		*ST3GAL5* (*SIAT9*)	Bladder	Exogenous GM3 inhibits proliferation, cell adhesion	[109]
	GM2	*B4GALNT1*	Breast	GM2 higher expression is associated with cancer cell stemness	[110]
		*B4GALNT1*	Ovarian	Retinoid-resistant ovarian cancer cells have higher level of GM2	[105]
		*B4GALNT1*	Lung cancer	Promotes metastasis and tumorigenicity in vivo	[111]
	GM1	*B3GALT4*	Colorectal	Appears upon anti-EpCAM-based inhibition of cell proliferation	[112]
	GD1α *	*ST6GALNAC3* *ST6GALNAC4* ***ST6GALNAC5*** *ST6GALNAC6*	Breast	Promotes brain metastasis and cell adhesion (ST6GALNAC5 in breast cancer cells and enhances their adhesion to brain endothelial cells)	[113]
	GD1b	***B3GALT4***	Breast	Exogenous and endogenous GD1b induces apoptosis in vitro	[114]
	GD2	***B4GALNT1***	Breast	Proliferation, cancer stemness	[95,115]
		*B4GALNT1*	Lung	Promotes proliferation and invasion	[98]
		*B4GALNT1*	Sarcoma	Enhancement of malignant properties	[96,116]
	GD3	*ST8SIA1*	Glioblastoma	Promotes proliferation and invasion	[117]
		***ST8SIA1***	Breast	Promotes migration/invasion, metastasis in vivo and cell adhesion	[81,83,115]
		*ST8SIA1*	Ovarian	Inhibits the antitumor NKT cell response	[118]

## Data Availability

MDPI Research Data Policies.

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
