# Peer review of "Deciphering the Importance of Glycosphingolipids on Cellular and Molecular Mechanisms Associated with Epithelial-to-Mesenchymal Transition in Cancer"

_biomolecules, 2021, doi:10.3390/biom11010062_

Round 1

Reviewer 1 Report

The authors tried to reviewed the current state-of-art of the role of GSLs in the multistep cancer-associated EMT process, this is definitely interesting topic. Unfortunately, I don´t think they succeeded.  The importance of GSLs in the cancer field, especially in the context of the anti-cancer therapy, massively expands every year, this is definitely true, but this review does not fulfil its own goal- to review the role of particular GSLs in the cancer-associated EMT process. It rather represents the overview of what is known about the GSLs in the context of particular cancer or cancer model, it says very little about the importance of particular GSL in the physiology of the cancer cell subpopulations, that undergo EMT or those cancer subpopulations displaying epithelial or mesenchymal features. There is another review, Russo et al, 2018 (PMID/30559216), describing almost the same (chapter GSL and cancer), and addressing much better the relationship between particular GSL metabolism during the EMT process. So, I don´t see the point of novelty here. The authors probably wanted to rescue this fact by adding the chapter 5 to the MS body, but this chapter itself rather stays apart than become valuable inner part of the draft. The chapter 4.1.2 is written in a quite confusing way too.

Minor comments: I would suggest not to use the word differentiation in the context of EMT (27). Figure 1 - Basement membrane degradation; Figure 2 - Lactose-ceramide should be replaced by Lactosylceramide; 121- "called as"

Reviewer 2 Report

This is a well written through review on glycosphingolipids in EMT.  Below are minor suggestions/corrections.

There have been a few recent publications examining levels of glycosphingolipids in liquid biopsies (plama/serum/etc) and the association of specific GSLs with cancer progression and aggressiveness.  These studies with others examining GLSLs in human cancer patients, highlighted in an additional subheading, would add clinical relevance/significance to many of the cell based studies.  

The section entitled "An overview of current methodologies to characterize GSLs" could also benefit from a sentence or two about the surge in untargeted metabolomics and lipidomics studies and their relevance to the study of GSLs.

Page 6 Line 201 and 202 lists colorectal and prostate cancer cell lines, but only CRC cell lines are named in the ().

Reviewer 3 Report

Authors reported the involvement of glycosphingolipids (GSLs) in epithelial- to-mesenchymal transition (EMT) in cancers, that is now considered to be a critical event for tumor progression and probably metastasis by collecting a number of related studies. This subject is quite important in clinical field, and also very interesting in the research of glycobiology. They tried to summarize reports in which GSLs are involved in not only EMT of cancers, but also broader aspects of cancer properties. As a new trial, this review has some significance. However, contents became largely diverse, and not deeply exploratory into roles of GSLs in EMT.

   Furthermore, there are a number of defects in their descriptions. Therefore, this paper is hard to be accepted for the publication at least in its present form.

Individual issues have been summarized as follows,

  1. Tumors they handled in this paper were largely deviated. Eventually, they mainly reported about some solid tumors such as breast cancers and ovarian cancers. More over, they did not clarified molecular mechanisms for the roles of GSLs in EMT. They just collected many phenomenological events.

  1. The authors reported results of engineering of glycosyltransferase genes such as GM3 synthase (p8). However, they did not describe about products of those enzymes, while there should be many kinds of GSL structures at the down-stream of the enzyme action point. The reviewer believes that we need to clarify the significance of each sugar structure, and to investigate function of each GSL by elucidating interaction between sugar chains and their recognizing molecules. They need to pay much more attention on this point.

  1. They frequently cited experimental data with exogenous GSLs. Addition of exogenous GSLs sometimes results in interesting outcome in target cells. It may some time help our standing of functions of native GSLs, but not necessarily correct.

  1. In Table 1, they summarize various references in which functions of GSLs were studied in not only EMT, but also in many other aspects. However, the contents are not well organized and not accurate. They collected some fragmental reports, and categorized in too simple manner. There should be much more reports on the roles of gangliosides in various cancers. Furthermore, their descriptions cited from ref.107, ref.108 seem to be seriously inaccurate. For example, they wrote that GM1 enhances “Cell proliferation” in “Colorectal cancers” citing ref.107. In fact, GM1 was related to the anti-cancer effects of anti-EpCAM monoclonal antibodies according to ref.107. More over, ref.108 was cited completely erroneously, since “GD1a” should be GD1alpha. In the cited paper, no function of GD1alpha (and of course of GD1a) had been analyzed at all.

Thus, I have to say that unfortunately, the authors descriptions seem not to be reliable.

  1. At last, they described about standard and advanced methods for characterization of GSLs. They need to mention their own way of approaches and directions in the future.

  1. Typographical errors are sometimes found, e.g. in Figure 1 degradation is degrdation.

Reviewer 4 Report

Reviewer comment

The author reviewed the role of Glycosphingolipids (GSLs) in tumorigenesis. They evaluated a large number of data indicating that several members of this family might contribute to epithelial-to-mesenchymal transition (EMT), a process involved in tumour progression, and the formation of new distant metastatic sites (mesenchymal-to-epithelial transition or MET).

I found the review informative, and citations appropriate. Data however are given as compilation of data, in many cases, giving gene names without explanation of their function in the given tumor . This makes reading a bit difficult for readers not familiar with the GSLs or the cancer type mentioned. I would have preferred to describe the role of GSLs with respect to each cancer type, as the final aim of the reviewers appears to which to demonstrate that GSLs might be important key factors in the EMT  and MET in several tumors. As it stands, it suggesting that they have a common function in tumorigenesis in all cancers. This might be a too symplicistic interpretation of given data.  

Minor points    

146-149  The prominent expression of Gb3 in epithelial cancers has led to various Gb3 targeting treatment attempts. For instance, knockdown of Gb3S (encoded by A4GALT) using siRNA in colon cancer epithelial cells [45] or using Shiga toxin in patient-derived gastric adenocarcinomas [51], decreased the cells capacity to migrate and to proliferate in vitro and in vivo compared to cells lacking Gb3

Is this phrase correct? ….

264 “Previous publication demonstrated that GM3 and GM1 were in favor of tumorigenicity”.= Previous publication demonstrated that GM3 and GM1 (overexpression?) were in favor of tumorigenicity.

Line 373 -374 “Our review demonstrates that GSLs actively modulate EMT through molecular and cellular processes in cancer cells.”

I do not think this conclusion can be made  . I would delete this phrase

Reviewer 5 Report

Interesting topic and the authors present a general summary about the role(s) of GSLs in the transition of epithelial to mesenchymal cells and back again in cancer. Discussion of unanswered questions in the conclusion section is helpful.

The topic as presented is very broad and dense to read. A major problem this reviewer had was the lack of discussion of steps indicating how GSL-induced alterations in signal transduction actually affected the transition. For example, see the discussion of SSEA4 and its effect on Claudin-7 and E-cadherin, need to indicate what Akt specifically affects to induce cell proliferation – those steps could be added to figure 3.  When discussing monosialoganagliosides (Section 4.1.1) in the second paragraph the authors say that GM3 and GM1 inhibited cell growth and have a pro-tumoral action. If they inhibit growth need to say how they have a pro-tumoral action, was it due to an E-M transition. Need explanations.

In the discussion of the effect of inhibiting synthesis of Glc-Cer using EtDO-P4 need to reiterate that it alters synthesis of all three classes of GSL. Need to include discussion of how changes in one GSL may be accompanied by changes in others which may also affect behavior. While this example brings that possibility to the fore, it is something that should be considered throughout.

Minor points:  

  1. Reference 88 is not to Battula et al as indicated it should be on lines 290, 310
  2. Need a list of abbreviations used. Also, use the same abbreviation when discussing an enzyme. An example of inconsistency is the use of ST3GAL5 initially and then shifting to GM3S.
  3. Line 300: GD1 should be GD1b
  4. Did not edit but there are instances where the wording is awkward

Round 2

Reviewer 1 Report

Dear authors,

I appreciate your effort to update your manuscript, however, I still don´t agree with your idea, that generalizes a very precise phenomenon of ´cancer-associated EMT´ to  "cell biological and molecular processes such as cell proliferation, (in-) dependent cell growth capacitie, as well as directed and random cell motility". Those and others (including EMT itself) are general hallmarks of carcinogenesis, and only when put to a detailed context of particular cancer cell subpopulation or to particular "phase of carcinogenesis", they can reveal their importance for EMT/MET.

Therefore, I would suggest to consider the change of the manuscript title (I would prefer the term carcinogenesis rather than EMT) and to summarize the particular roles of GSLs during EMT/MET processes in the special chapter, which will describe the effect of GSL on morphology/migration/invasion in cancer cells subpopulation bearing either epithelial (E), mesenchymal (M) or hybrid (E/M) features or occuring during e.g. cancer dissemination.

Parts bellow are perfect examples what I mean:

Cheung et al., demonstrated that elevated SSEA3 in CD44+/CD24-breast
cancer cells leads to enhanced tumor growth in vivo and therefore suggested SSEA3 as a breast cancer stem cell marker [40].

or

They demonstrated a muscle invasive cell line UMUC3 with
elevated GD2 expression showing cancer stem cell property (CD44+/CD24-) as well as a mesenchymal phenotype. Moreover, the same GD2+ cells displayed higher expression for vimentin and lower Ecadherin expression than GD2- cells [95]. Battula et al., observed a similar pattern in GD2 positive
breast cancer stem cells with association to CD44+ and CD24- stem cell markers and generation of spheroids in vitro [91] .

Author Response

Please see response in attached PDF file.

Reviewer 5 Report

The authors addressed most of the comments made by this reviewer but missed a few. They neglected to address the comment that when discussing monosialoganagliosides (2nd paragraph, Section 4.1.1) the authors say that GM3 and GM1 inhibited cell growth and have a pro-tumoral action. If they inhibit growth need to say how they have a pro-tumoral action, was it due to an E-M transition. Also, should address the question of what GSL changes addition of a GSL might have on the expression of others in a given experiment.

As stated previously, GD1 needs to read GD1b (lines 314 and 326) as GD1a is derived from GM3 and GD1b is the only disialoganglioside derived from GD3. In Figure 2 and on lines 121 and 122 the Greek letters a and ß should read a and b as they are the subscripts used for the gangliosides shown/discussed. GD1a actually refers to a different molecule than the one shown in figure 2.

These comments should be addressed prior to acceptance.

Author Response

(The authors gave the same response as above.)
